# Toward conformational identification of molecules in 2D and 3D self-assemblies on surfaces

Ali Hamadeh[1], Frank Palmino[1], Jérémie Mathurin[2], Ariane Deniset-Besseau [2], Louis Grosnit[1], Vincent Luzet[1], Judicaël Jeannoutot[1], Alexandre Dazzi[2] & Frédéric Chérioux [1✉]

The design of supramolecular networks based on organic molecules deposited on surfaces, is highly attractive for various applications. One of the remaining challenges is the expansion of monolayers to well-ordered multilayers in order to enhance the functionality and complexity of self-assemblies. In this study, we present an assessment of molecular conformation from 2D to 3D supramolecular networks adsorbed onto a HOPG surface under ambient conditions utilizing a combination of scanning probe microscopies and atomic force microscopy- infrared (AFM-IR). We have observed that the infrared (IR) spectra of the designed molecules vary from layer to layer due to the modifications in the dihedral angle between the C=O group and the neighboring phenyl ring, especially in the case of a 3D supramolecular network consisting of multiple layers of molecules.

[1] Université de Franche-Comté, FEMTO-ST, CNRS, F-25000 Besançon, France. [2] Université de Paris-Saclay, Institut de Chimie-Physique, F-91400 Orsay, France. ✉email: frederic.cherioux@femto-st.fr

On-surface self-assembly involves the bottom-up construction of nanostructures based on molecular building blocks adsorbed on a surface. This approach has seen successful development over the three past decades[1–4]. However, one of the significant remaining challenges in this research field is the transition from 2D to 3D nanostructures[5,6]. Such a shift holds the potential to enable numerous organic-based devices in electronics, optoelectronics, or spintronics. Despite notable progress, the mastery of 3D self-organization remains incomplete. Achieving this critical goal requires two primary challenges: (i) controlling molecule-molecule interaction out-of-the-plane of the underlying surface and (ii) developing accurate analytical tools capable of non-invasively probing through the thickness of materials at the nanoscale. The investigation of molecule-molecule interactions beyond the surface plane is being explored through the use of 3D molecular blocks[7–9] or controlled growth of multilayers composed of planar molecules[10–12]. Yet, these 3D structures often retain surface-like characteristics, extending only to a limited depth, generally bi- or three-layer thickness. The restriction in 3D extension primarily stems from an incomplete understanding of growth mechanisms transitioning from 2D to 3D, particularly under ambient conditions. Regarding analytical tool development, several prominent surface characterization techniques[13] such as Scanning Tunneling Microscopy (STM) or Atomic Force Microscopy (AFM), offer new insights into on-surface self-assembly. Both are non-destructive analytical instruments. Nevertheless, these methods possess limited chemical sensitivity and are largely constrained to investigating within the surface plane. These experimental constraints hinder their development and broad adoption across various communities, thereby reducing the overall impact of on-surface 3D self-assembly. Polarization modulation-infrared reflection-adsorption spectroscopy (PM-IRRAS) stands out as a valuable tool for characterizing thin films or monolayers adsorbed onto metal surfaces due to its high surface sensitivity[14]. Nonetheless, while PM-IRRAS provides insight into the conformational organization of molecules in a monolayer, it lacks the high spatial resolution of scanning probe microscopies. In response, infrared (IR) and Raman nanospectroscopy, coupled with AFM, have undergone significant enhancement in the past decade[15,16]. Tip-Enhanced Raman Spectroscopy (TERS)[17] combines AFM and visible light for nanoscale Raman spectroscopy, but it relies on the tip properties and is challenging to implement. AFM-IR, in contrast, aims to combine AFM's high spatial resolution with IR spectroscopy's chemical identification capabilities[18–20]. Additionally, AFM-IR can probe layers in depth, making it a promising tool for studying 3D self-assemblies on surfaces. However, it's worth noting that in the literature, there are only a few reports of AFM-IR studies conducted on few-layered samples, specifically those with a thickness of a few nanometers[21–23]. Investigating monolayers remains a significant challenge but possible here due to the specifications of our AFM setup. Nevertheless, AFM-IR's lateral resolution currently hinders the determination of supramolecular assembly patterns if their periodicity is smaller than 25 nm[21].

To address this, we have developed an effective approach combining STM under ultra-high vacuum, AFM, and AFM-IR under ambient conditions. This approach enables the exploration of supramolecular self-assembly resulting from the deposition of precisely designed molecules on a Highly Ordered Pyrolytic Graphite (HOPG) surface. Our aim is to definitively address the question: "How can we determine the conformation of molecules within a 2D monolayer or a 3D multilayered supramolecular network adsorbed onto a surface?" This approach not only identifies the adsorption model of molecules on the HOPG surface and the entire 2D supramolecular organization within the surface plane but also delves into the 3D organization of multi-layered supramolecular assemblies by deeply investigating the IR fingerprint of the molecules. This method holds a general applicability and has the potential to significantly enhance our understanding of molecular conformation during self-assembly on surfaces.

## Results

**AFM on a monolayer**. The selection of Highly Oriented Pyrolytic Graphite (HOPG) as a model for an air-stable crystalline conductive surface for creating extended 2D supramolecular networks has already been demonstrated[24]. Our molecular design incorporates two crucial components: (i) a carbonyl functional group serving as an IR tag and (ii) C18 alkyl chains to facilitate the formation of 2D or 3D extended supramolecular network on a HOPG surface guided by the subtle balance of molecule-surface and molecule-molecule interactions. We synthesized an ester derivative (octadecyl 4'-octadecyloxy-4-biphenylcarboxyloate, EsterOC18, Fig. 1a), based on a biphenyl core surrounded by two n-octadecyl chains, one linked to the aromatic core through an ester function. The side chain length of the EsterOC18 molecule measures 5.12 nm (Fig. 1a).

Figure 1b depicts a high-resolution STM image, acquired at room temperature (RT) under ultra-high vacuum (UHV) of the EsterOC18/HOPG interface. This interface was obtained by the deposition of a solution of EsterOC18 molecules (Dichloromethane, $5 \cdot 10^{-5}$ mol·L$^{-1}$) on a HOPG surface through spin-coating under ambient conditions (see method for the detailed procedure). Subsequently, the resulting substrate was introduced to UHV. Within the STM image, a well-defined compact periodic network constituted by bright lines separated by darker strips is observed. The periodicity between the bright lines is $5.60 \pm 0.1$ nm (Fig. S1a, b). Notably, leveraging the high resolution of STM, we discern that the darker stripes consist of two parallel nanorods, each measuring $2.1 \pm 0.2$ nm in length (white double arrows, Fig. 1b). These nanorods align within the same row and exhibit a 120° rotation from one dark stripe to the next.

Next, another sample was prepared by deposing a solution of EsterOC18 molecules (Dichloromethane, $1 \cdot 10^{-5}$ mol·L$^{-1}$) onto a HOPG surface via spin-coating (see method for a detailed procedure). This sample was investigated using atomic force microscopy (AFM) under ambient conditions. The AFM topography images (Fig. 1c) reveal the existence of compact 2D domains, consistent with what was observed in STM images (Fig. 1b). These two-dimensional domains showcase well-arranged bright stripes segregated by darker stripes. As depicted in Fig. S1, the periodicity between the bright line stripes remains constant at $5.60 \pm 0.1$ nm. The thickness of the two-dimensional domains was evaluated by analyzing the AFM topography of a HOPG surface coated with a sub-monolayer of EsterOC18 molecules, resulting in a thickness of $0.8 \pm 0.05$ nm (Fig. S2). Subsequently, the same sample was introduced into UHV and imaged using STM. Following this, it was taken out of the UHV setup and analyzed again using AFM under ambient conditions. In both STM and AFM images, 2D domains exhibit well-structured bright stripes interspersed with darker stripes, maintaining a consistent distance of $5.60 \pm 0.1$ nm. This confirms that the networks remain unaltered by the imaging conditions, whether UHV or ambient conditions.

**AFM on multilayers**. Figure 2a shows a topography AFM image ($2 \times 2 \, \mu m^2$) recorded after the deposition of a concentrated solution of EsterOC18 molecules (Dichloromethane, $1 \cdot 10^{-3}$ mol·L$^{-1}$) by spin-coating on a HOPG surface (see method for a detailed procedure). The formation of large compact islands

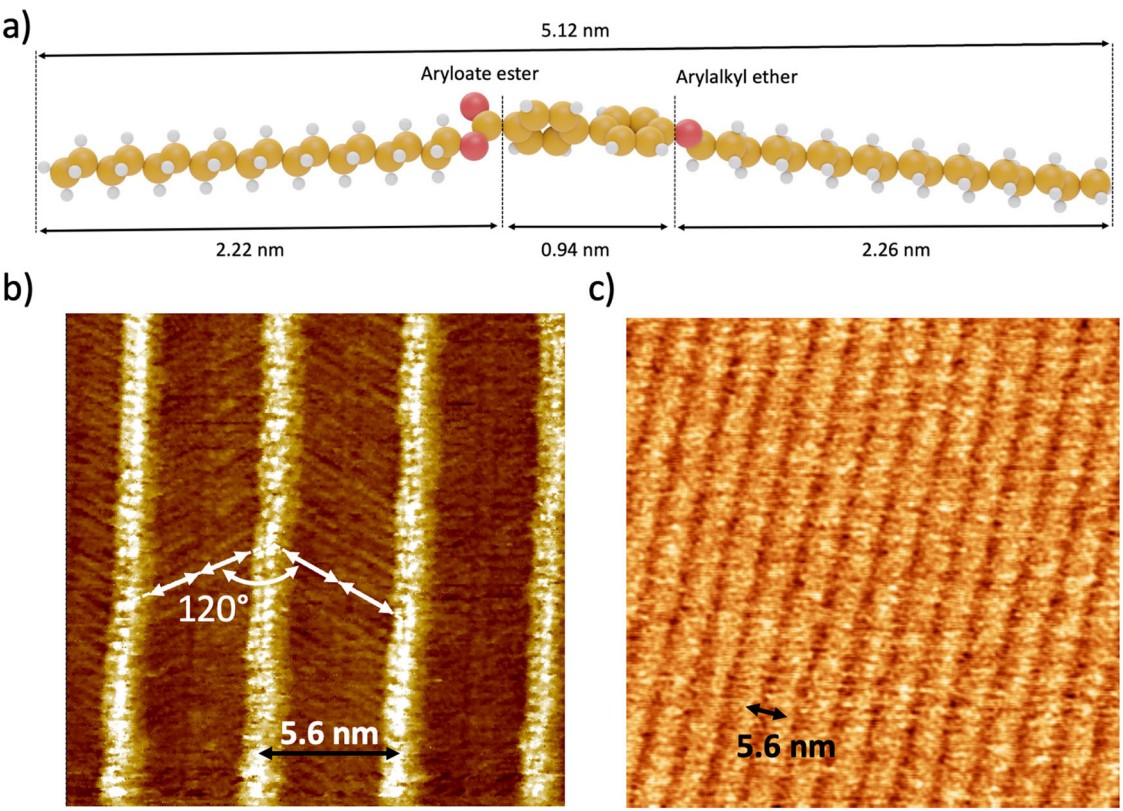

**Fig. 1 STM and AFM images of monolayer of EsterOC18 molecules on a HOPG surface. a** Structural model of octadecyl 4'-octadecyloxy-4-biphenylcarboxyloate (EsterOC18) Oxygen atoms: red, Carbon atoms: yellow, Hydrogen Atoms: grey. **b** STM image ($20 \times 20$ nm$^2$, $V_s = -1.3$ V, $I_t = 20$ pA, T = RT) of a monolayer of EsterOC18 molecules deposited on a HOPG surface. The dark nanorod (blank double arrows) are rotated by 120° between two consecutive dark stripes. **c** Topography AFM image ($65 \times 65$ nm$^2$). AFM and STM images show a compact periodic network constituted by bright lines separated by darker stripes, with a periodicity of $5.60 \pm 0.1$ nm, highlighted by the black double arrows in (**b**) and (**c**).

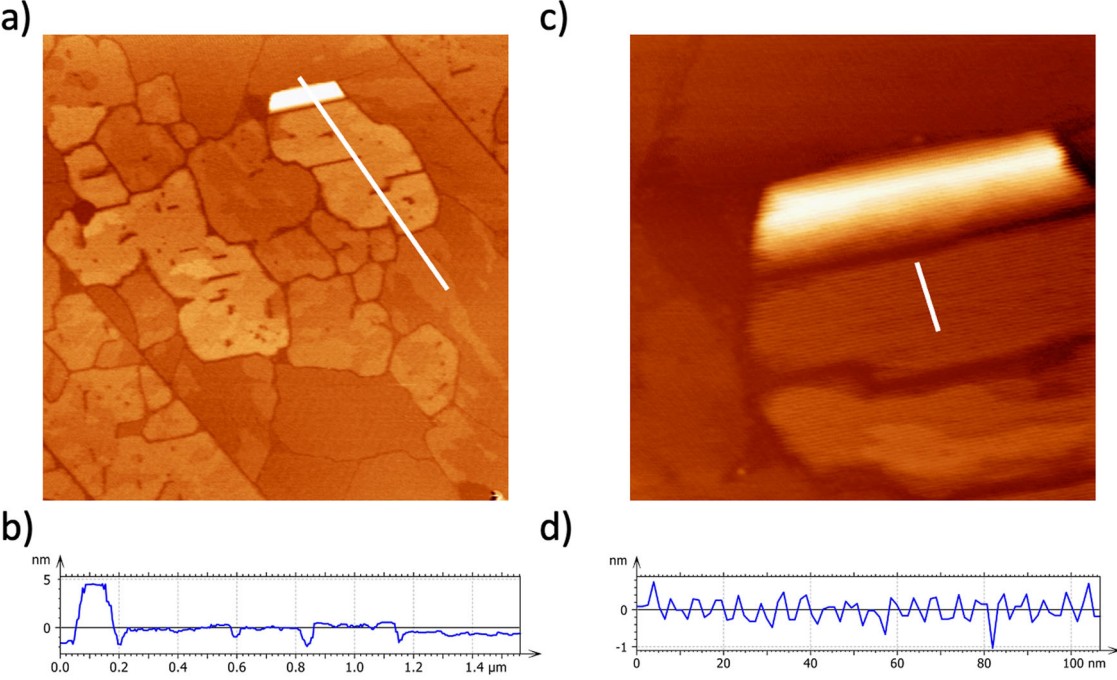

**Fig. 2 AFM images of multilayers of EsterOC18 molecules adsorbed on a HOPG surface. a** Large-scale ($2 \times 2$ µm$^2$) topography AFM of multilayers of EsterOC18 molecules deposited on a HOPG surface. **b** Z-profile taken along the white line of showing the formation of islands with different apparent height. **c** Zoom ($600 \times 600$ nm$^2$) over multilayered islands are constituted of bright lines separated by darker stripes. **d** Z-profile taken along the white line of (**c**).

is observed (Fig. 2a). The thickness of these islands varies from 0.7 to 6 nm (Fig. 2b). A deeper investigation of the topography AFM image shows that the 3D compact islands are still constituted by alternating bright and dark lines, with the previously observed periodicity of $5.60 \pm 0.1$ nm, irrespective of their thickness (Fig. 2c).

**AFM-IR on multilayers**. The IR fingerprint of the compact islands, in relation to their thickness, was subsequently investigated through AFM-IR experiments. AFM-IR absorption spectra were recorded on another HOPG substrate after the deposition of a solution of EsterOC18 molecules (Dichloromethane, $1 \cdot 10^{-3}$ mol·L$^{-1}$, see method for detailed procedure). The existence of 3D compact islands with varying thicknesses (ranging from monolayer to 4 layers) on the substrate was confirmed by obtaining AFM topography images (Fig. 3a). Subsequently, AFM-IR absorption spectra were captured from compact islands with differing thicknesses.

Specifically, AFM-IR absorption spectra were collected from the 2D monolayer (blue cross, Fig. 3a) and the 3D multilayers (red cross, Fig. 3a). To facilitate comparison with the AFM-IR absorption spectra, the FT-IR spectrum of a powder of EsterOC18 molecules in the pure solid state has been recorded (bulk, black dashed line in Figs. 3 and S3). As evident in Fig. 3b, c, the absorption spectrum of the multilayers (red line) closely resembles that of the bulk (black dashed line). The anticipated features for EsterOC18 are all present: C=O stretching at 1703 cm$^{-1}$, C=C stretching at 1602 cm$^{-1}$, and 1580 cm$^{-1}$ (Fig. 3b, c). Notably, the primary distinction between the two spectra lies in the weak, broad band centered at 1740 cm$^{-1}$, which is evident in the multilayer spectrum but absent from the solid spectrum (Fig. 3c).

Conversely, for the monolayer scenario, the AFM-IR spectrum displays significant dissimilarity. The C=O bond stretching experiences a shift to 1740 cm$^{-1}$ (depicted by the blue line in Fig. 3c) and exhibits greater broadness compared to the solid state (represented by the black dashed line in Fig. 3c). Additionally, no C=C stretching is observed at 1602 cm$^{-1}$ and 1580 cm$^{-1}$ in the monolayer spectrum.

The chemical mapping of the area depicted in Fig. 3a was conducted using a quantum cascade laser (QCL) tuned to the fixed wavenumbers 1600 cm$^{-1}$ (Fig. 3d), 1710 cm$^{-1}$ (Fig. 3e), and 1730 cm$^{-1}$ (Fig. 3f). In all instances, the signal intensity is evenly distributed across the surface of the islands with the same thickness, as highlighted by adsorption spectra collected at various points within the multilayered network area (Fig. 3). Specifically, within the chemical mapping at 1600 cm$^{-1}$ corresponding to the C=C bond stretching, the signal is virtually absent (in blue, Fig. 3d) in the region attributed to the monolayer. Conversely, a strong signal is detected in the area corresponding to the multilayered network (illustrated in red, Fig. 3d). For the band centered at 1710 cm$^{-1}$, the highest signal intensity is situated over the multilayer, while for the 1730 cm$^{-1}$ band, the maximum signal intensity is observed in the monolayer zone (as illustrated, respectively in Fig. 3e, f).

## Discussion

Close examination of STM and AFM images reveals the consistent periodicity exhibited by the supramolecular networks formed after depositing EsterOC18 molecules onto a graphite surface, regardless of whether they take the form of 3D islands or 2D monolayers. Leveraging the high-resolution capabilities of STM images, we were able to observe the spatial arrangement of aliphatic chains surrounding the EsterOC18 molecules, resulting

in the development of an adsorption model describing the 2D monolayers. In the monolayer scenario, the experimental periodicity ($5.60 \pm 0.1$ nm) closely matches the length of EsterOC18 molecules (5.12 nm), suggesting the absence of lateral n-octadecyl chain interdigitation, as demonstrated in the STM images (depicted in Fig. 1b). Furthermore, since the nanorods are rotated at 120° between adjacent dark stripes (as shown in Fig. 1b), we infer that EsterOC18 molecules align along the <100> direction of the HOPG surface, forming a densely packed 2D lamellae of parallel-aligned, straight n-octadecyl chains, aligning with the Groszek model (Fig. 4a)[25]. Importantly, the identical periodicity observed in both 2D and 3D layers enabled us to extend the proposed 2D organizational model to elucidate the arrangement of 3D islands.

Furthermore, AFM-IR signatures provide additional insightful details compared to STM and AFM images. Notably, the infrared signature of monolayers displays distinct differences when compared to that of multilayers or bulk solids. This divergence in the IR signatures furnishes essential insights for understanding the unique attributes and characteristics of supramolecular structures in different dimensions. AFM-IR exhibits high sensitivity to the incident polarization of the IR light[26,27]. In our setup the incident light has an angle about 30° from the surface and is polarized in the plane of incidence. Considering the gold coating of the tip and this configuration of illumination, we have already demonstrated that the electric field below the tip is dominated by its perpendicular component[26]. Consequently, for functional groups situated in the plane of the underlying HOPG surface, the dipole's cross-section of these functional groups and the incident electric field is nearly zero, causing their IR fingerprint to cancel out. Utilizing this phenomenon, it becomes feasible to determine the relative configuration of functional groups comprising EsterOC18 molecules within supramolecular self-assemblies on a HOPG surface.

In the AFM-IR spectrum of a monolayer, the absence of C=C stretching peaks at 1602 cm$^{-1}$ and 1580 cm$^{-1}$, present in the bulk, provides substantial evidence that the aromatic rings align parallel to the HOPG surface plane. This observation indicates that the aromatic rings within the monolayer adopt an in-plane orientation concerning the substrate. Notably, within our research framework, we observed a significant phenomenon related to C=O stretching in the EsterOC18 monolayer. The C=O stretching band appears at 1740 cm$^{-1}$, showcasing a hypsochromic shift of 37 cm$^{-1}$ from the C=O stretching band in the FT-IR spectrum of the pure solid (bulk). This shift holds particular interest, as it signifies the absence of electronic conjugation between the C=O group and the surrounding phenyl ring. These collective findings provide compelling evidence that the C=O bond within the EsterOC18 monolayer is oriented out of the HOPG surface plane and distinct from the neighboring phenyl ring. With these features in mind, the adsorption model corresponding to the monolayer of EsterOC18 molecules adsorbed on a HOPG surface is illustrated in Fig. 4a, b. This model effectively accounts for all experimental data, with the phenyl rings aligned parallel to the HOPG surface due to molecule-surface interactions, while the C=O groups reside out of the surface plane (depicted in Fig. 4b).

In the case of multilayers, the band associated with C=C bond stretching is now observed at 1602 cm$^{-1}$, similar to the bulk (depicted in Fig. 3c). However, two bands related to C=O bond stretching are detected. One is situated at 1702 cm$^{-1}$, identical to the bulk spectrum, while the second, with a weaker intensity, appears at 1740 cm$^{-1}$, resembling the AFM-IR spectrum of the monolayer. Consequently, it can be inferred that two orientations or conformations of EsterOC18 molecules comprise the multilayered networks (depicted in Fig. 4c).

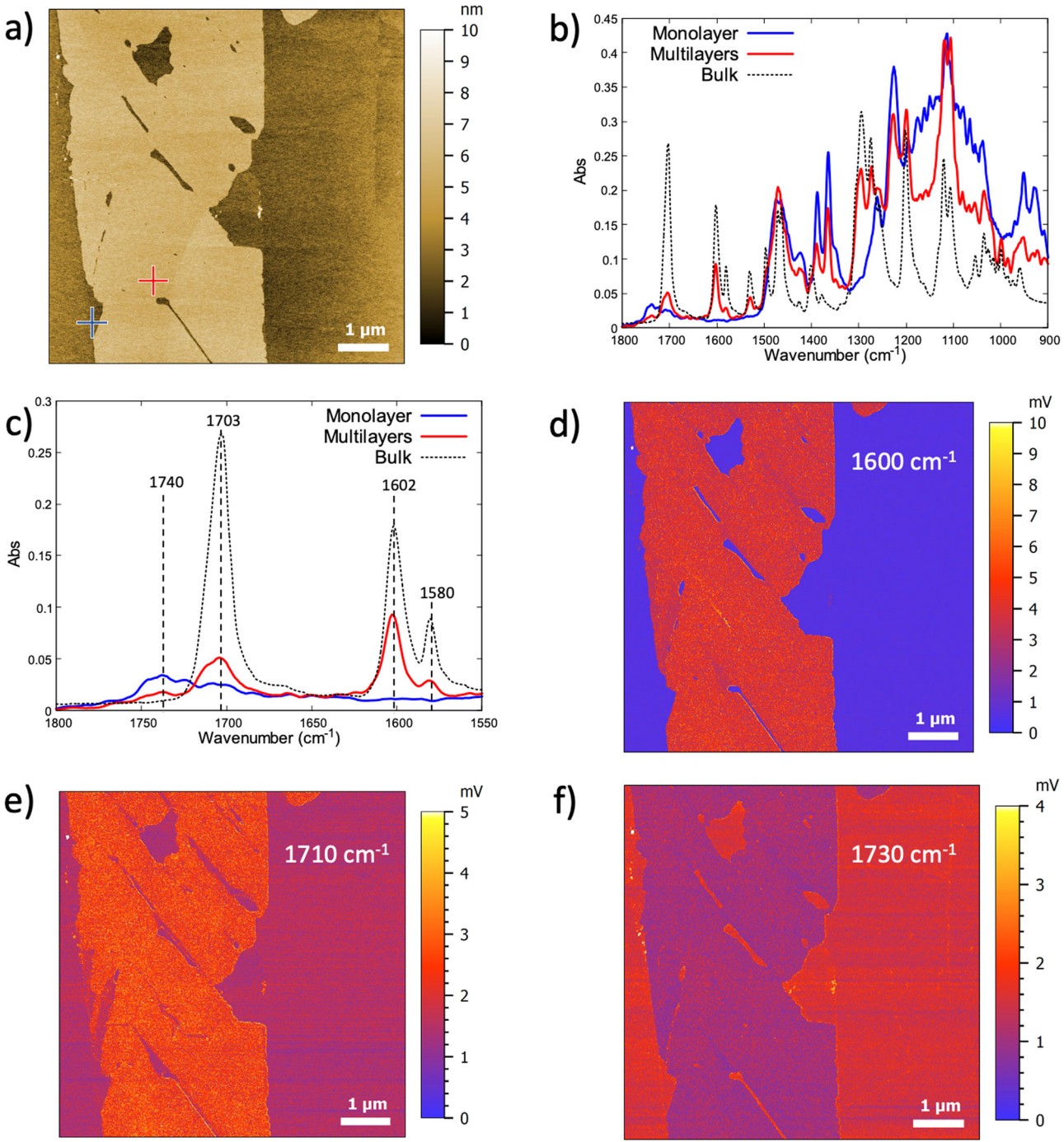

**Fig. 3 AFM-IR spectra and mapping of EsterOC18 adsorbed on HOPG surface. a** Large-scale ($7 \times 7\,\mu m^2$) topography AFM of multilayers (thickness of 3 nm) and a monolayer of EsterOC18 on a HOPG surface. **b** AFM-IR absorption spectra recorded for multilayers (red line) and a monolayer (blue line), respectively on the red and the blue cross of (**a**). The FT-IR spectrum of a pure powder of EsterOC18 is represented by the black dashed line (**c**) Zoom of (**b**) in the C=O and C=C bonds stretching region (1550–1800 cm$^{-1}$). AFM-IR mapping corresponding to the topography shown in Fig. 3a recorded at (**d**) 1600 cm$^{-1}$, (**e**) 1710 cm$^{-1}$ and, (**f**) 1730 cm$^{-1}$. At 1600 cm$^{-1}$ and at 1710 cm$^{-1}$, the intensity of the AFM-IR signal is stronger on the multilayers than on the monolayer. At 1730 cm$^{-1}$, the intensity of the AFM-IR signal is stronger on the monolayer than on the multilayers. The scale color of the working function is displayed.

The first layer, closest to the surface (highlighted in yellow in Fig. 4b), assumes the same conformation as the one constituting the monolayer, attributed to the molecule-surface interaction (Fig. 4c). The corresponding C=O bond stretching occurs at 1740 cm$^{-1}$. In contrast, the conformation of EsterOC18 molecules in the upper layers of a multilayer system (illustrated in blue in Fig. 4c) differs significantly. The phenyl rings rotate to enhance

their conjugation with the C=O group, resulting in the shift of the C=O bond stretching to 1702 cm$^{-1}$, akin to the bulk (depicted in Fig. 4d). As the phenyl rings are now out of the HOPG surface plane, the C=C stretching bonds become observable in the AFM-IR spectrum.

In the monolayer, the band linked to C=O bond stretching is notably broad (ranging from 1700 to 1750 cm$^{-1}$), and within the

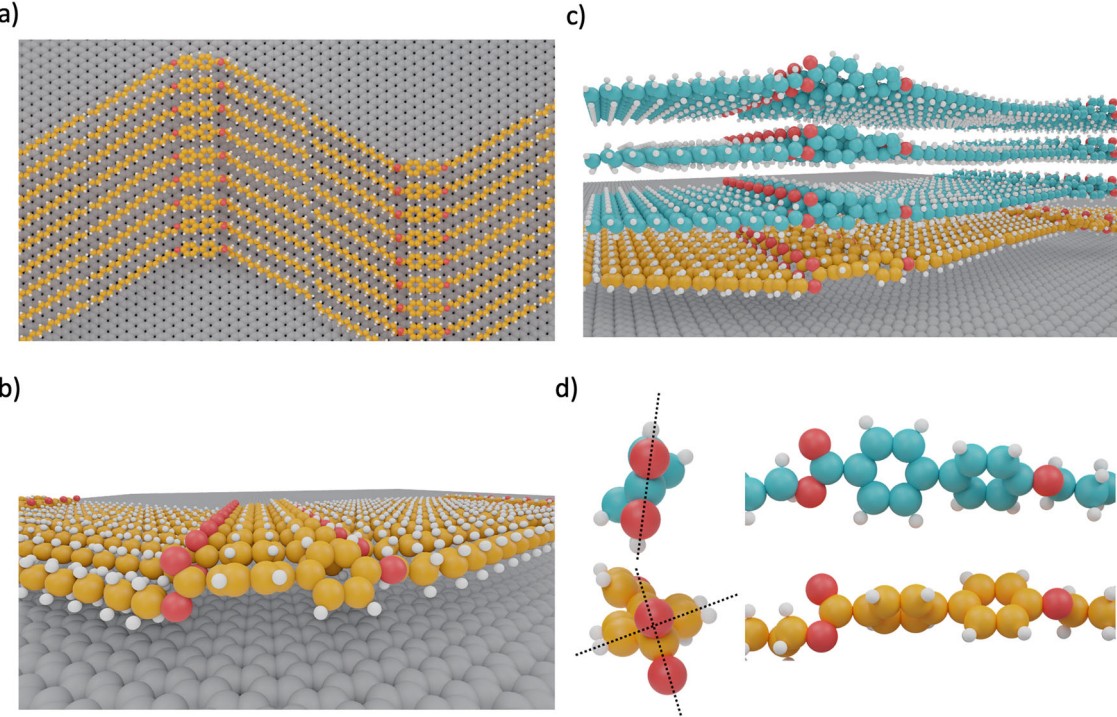

**Fig. 4 Adsorption models of EsterOC18 on HOPG surface. a** Top view of the proposed molecular model for a monolayer of EsterOC18. **b** Side view of (**a**). The phenyl ring linked to the ester function is out-of-the-plane of the ester function and in-the-plane of the HOPG surface. **c** Side view of the proposed molecular model of a 4-layered EsterOC18 supramolecular network. The carbon atoms of the first adlayer are painted in yellow, while the carbon atoms of the other layer are painted in blue. **d** Two side-views of EsterOC18 molecules involved in the first layer (carbon atoms painted in yellow) and in other layers (carbon atoms painted in blue) of a multilayered supramolecular network. The n-octadecyl chains have been omitted for clarity. In the layer close to the HOPG surface, the phenyl ring is still out-of-the-plane of the neighboring ester function and in-the-plane of the HOPG surface. In the other layers, the phenyl ring is now in-of-the-plane of the neighboring ester function and out-of-the-plane of the HOPG surface. The conformational difference of EsterOC18 molecules is highlighted by dashed black lines.

region covered by the EsterOC18 multilayer, the signal intensity of C=C stretching bonds and C=O bonds fluctuates by a factor of two between minimum and maximum intensity (Fig. S4). This observation leads to the assumption that the angle between the phenyl rings of EsterOC18 molecules and the HOPG surface, as well as the dihedral angle between the C=O group and the adjacent phenyl ring, is not fixed but varies from 0 to 90°. These variations arise from the modulation of the molecule-surface interaction based on the distance between the molecules and the HOPG surface. This finding underscores the sensitivity of AFM-IR in identifying these subtle differences in molecular conformation within mono- or multilayered supramolecular networks adsorbed on a surface.

## Conclusion

This study has showcased the high sensitivity and non-invasiveness of AFM-IR as a potent technique for exploring molecular conformation within supramolecular networks. By leveraging IR fingerprints obtained through AFM-IR experiments, we have highlighted the variation of the dihedral angle between the C = O group and the adjacent phenyl ring, within 2D to 3D supramolecular networks adsorbed onto a HOPG surface under ambient conditions. Notably, in the context of a 3D supramolecular network formed by stacked molecular layers, the conformation of molecules varies from one layer to another. The layer proximate to the substrate mirrors the conformation of the corresponding 2D monolayer, while the others closely resemble the bulk conformation. This study has successfully demonstrated that the amalgamation of STM, AFM, and AFM-IR not only elucidates the orientation and arrangement of surface-adsorbed

molecules but also discerns the molecular conformation across different layers of 3D multilayered supramolecular networks. This combined analytical approach could be universally applied to unravel the intricacies of on-surface supramolecular self-assemblies.

## Methods

**Synthesis of molecule.** All reagents were purchased from TCI, and used as received. The deuterated NMR solvents were purchased from Euriso-top. The NMR spectra were recorded using a Bruker AC-300 MHz spectrometer. FT-IR spectrum was recorded using Spectrum Two FT-IR spectrometer from PerkinElmer (ATR, transmission and absorption).

4'-hydroxy-4-biphenylcarboxylic acid (1 g, 42 mmol) was dissolved in dimethylformamide (10 mL) at room temperature. Then NaH (112 mg, 42 mmol) was added. Then 1-bromooctadecane was added (6.22 g, 168 mmol, in 40 mL of dimethylformamide solution) was added for 30 min (Fig. S5). The mixture is heated at 80° for 15 h. The grey solid is filtered and washed three times with 50 mL of water. The pure octadecyl 4'-octadecyloxy-4-biphenylcarboxyloate is obtained as a white solid (Yield: 65%). $^1$H and $^{13}$C NMR spectra are shown in Figs. S6 and S7, respectively. FT-IR spectrum is described in Fig. S3.

$^1$H RMN (300 MHz, CDCl$_3$) δ = 8.07 (d, J = 8.2, 4H), 7.61 (d, J = 8.2, 4H), 7.55 (d, J = 8.2, 4H), 6.98 (d, J = 8.2, 4H), 4.32 (t, J = 6,4, 2H), 4.00 (t, J = 6,4, 2H), 1.79 (h, J = 6.4, 4H), 1.26 (m, 60H), 0.88 (t, J = 6.4, 6H). $^{13}$C RMN (75 MHz, CDCl$_3$) δ = 166.69, 159.42, 145.19, 132.22, 130.06, 128.31, 126.41, 114.93, 68.16, 65.11, 31.94, 29.71, 29.67, 29.61, 29.56, 29.41, 29.38, 29.32, 29.27, 28.77, 26.08, 26.06, 22.70.

**Preparation of sample for STM imaging**. The experimental imaging was performed using a set-up of Variable temperature Scanning tunneling microscope (VT-STM) under ultra-high vacuum (UHV) conditions with base pressure inside the setup $7.2 \times 10^{-11}$ mbar, all STM-acquired images at room temperature using a tungsten tip using constant current mode. A 20 µl of EsterOC18 solution in dichloromethane with a concentration of $1 \cdot 10^{-5}$ mol·L$^{-1}$ was deposited onto the HOPG surface and the spin coating was carried out for 30 s with an angular velocity of 300 rpm. After spin coating, the sample was thoroughly dried at 80° C for 30 min in a vacuum oven Memmert Heater in order to evaporate the solvent from the HOPG substrate.

**Preparation of monolayer for AFM imaging**. The experimental imaging was performed at ambient conditions and monitored by AFM with Peak Force Mode by Bruker Icon using a super sharp probe (SSB) tip. A 20 µl of EsterOC18 solution with a concentration of $5 \cdot 10^{-5}$ mol·L$^{-1}$ was deposited onto the HOPG surface and the spin coating was carried out for 20 s with an angular velocity of 300 rpm. After spin coating, the sample was thoroughly dried at 80 °C for 30 min in a vacuum oven Memmert Heater in order to evaporate dichloromethane from the substrate.

**Preparation of multilayers for AFM imaging and AFM-IR imaging**. The experimental imaging was performed at ambient conditions and monitored by AFM with Peak Force Mode by Bruker Icon using a super sharp probe (SSB) tip. A 20 µl of EsterOC18 solution with a concentration of $1 \cdot 10^{-3}$ mol·L$^{-1}$ was deposited onto the HOPG surface and the spin coating was carried out for 20 seconds with an angular velocity of 500 rpm. After spin coating, the sample was thoroughly dried at 80 °C for 30 min in a vacuum oven Memmert Heater in order to evaporate dichloromethane from the substrate.

**AFM-IR experiment**. The AFM-IR system was an Icon-IR from Bruker using a multi-chip Quantum Cascade Laser (QCL, Daylight Solutions) covering the mid-IR range from 900 cm$^{-1}$ up to 1900 cm$^{-1}$. In this study, AFM-IR was used in tapping AFM-IR mode with a gold-coated silicon AFM tip (Nanoandmore PPP-NCHAu-MB-10). The Drive frequency was around 1680 kHz and the IR detection was around 272 kHz. To compensate the topography or mechanical change, PLL (Phase Loop Lock) has been activated for IR channel detection. The IR mapping acquisitions were made using a 0.3 Hz scan rate with a laser power of around 1–3% with a pulse width of 100 ns. All spectra are recorded after IR maps to select relevant regions of interest considering the chemical distribution. Due to the laser illumination configuration and the gold coating of tips, the polarization of the IR light is mainly perpendicular to the surface of the sample.

## Data availability

Any relevant data are available from the authors upon reasonable request.

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

## Acknowledgements

The authors acknowledge the financial support from the French National Research Agency through contract ATOMICHEM (ANR-22-CE42-0006) and from the Pays de Montbéliard Agglomération. A.D., A.D.-B., and J.M. acknowledge funding from Paris Ile-de-France Region-DIM "Matériaux anciens et patrimoniaux".

## Author contributions

V.L. and L.G. synthesized the molecules. A.H., J.J. and F.P. made A.F.M. and S.T.M. experiments. A.D., A.D.-B., and J.M. made the AFM-IR experiments. F.C., F.P., and A.D. designed all experiments and wrote the article.

## Competing interests

The authors declare no competing interest.

**Additional information**

