## [Peer Review File · Communications Chemistry]

Reviewers' comments:

Reviewer #1 (Remarks to the Author):

Hamadeh et. al. reported the conformation assessment of molecular self-assemblies from 2D to 3D. The AFM work and data analyzing are complete, but the mechanism remains elusive. The combination of SPM and AFM-IR is unique, however the manuscript may need to re-organize in an easy-to-read fashion.

1.If STM can evidence layered structures for this system?

2.If there is a strong computer simulation/calculation to support the mechanism?

3.“This combined analytical approach can be universally applied to unravel the intricacies of on-surface synthesis processes”: I am confusing regarding this conclusion; I think there is no evidence to demonstrate this.

Reviewer #2 (Remarks to the Author):

In “Toward Conformational Identification of Self-Assembled Molecules from 2D to 3D” Chérioux and colleagues, use atomic resolution STM and nanoscale resolution AFM-IR (an AFM-based analogue of IR spectroscopy) to study the intramolecular conformation of molecule-based supramolecular networks. The samples studied range from monolayers (characterized by a 2D arrangement/periodicity) and multilayers, which could in principle be considered 3D, as the authors did.

I like the manuscript approach which leverages the synergy (I believe for the first time) between atomic resolution STM and nanoscale resolution AFM-IR to evince the conformation of the biphenyl core and ester group in EsterC18 (octadecyl 4'-octadecyloxy-4-biphenylcarboxylate) on HOPG. The main claim of the paper is the qualitative determination of the conformation of the dihedral angle between the two aromatic groups in the biphenyl core and between one of the aromatic groups and the ester moiety. Specifically, in the monolayer and in the first layer of a multilayer sample, the biphenyl group is mostly in plane while the ester group is rotated (weak electronic conjugation) while in the successive layers the biphenyl assumes a conformation more akin its bulk form (more rotated) and the dihedral angle with ester group is reduced to increase the electronic conjugation between the 2 chemical moieties.

Overall, I find the manuscript novel, well written and of appropriate length. I also believe that it will appeal to a broad audience since measuring the conformation of molecules at the nanoscale is of interest in applications ranging from biology to molecular electronics. Technically, the manuscript is also relevant because although AFM-IR has recently increased in popularity, its application to very thin (i.e. monolayer samples) has been very limited due to the typical low signal intensity on such samples. While I'm generally enthusiastic about the manuscript, there are some inconsistencies between Figure 3, its caption, and the main text which contradict the main conclusions of the paper. (Perhaps the wrong figure 3 was used?) I would be happy to recommend the manuscript for publication if the authors can address this issue along with other minor suggestions detailed below.

1) I find the title and abstract a bit misleading as they seem to point to a sort of dynamic effect, while instead the “evolution of molecular conformation” refers (figuratively) to different samples, i.e. no evolution at all. Perhaps something like “assessing the conformation of molecules self-assembled in 2D

and 3D” or something like that may be better.

2) Since the molecules under study are characterized by many dihedral angles, i.e., conformations (both at the core and in the aliphatic chains) I think it would be useful to the reader to mention in the abstract what kind of molecules and which dihedral angles are analyzed. Otherwise, it is not clear what the main focus of the paper is before reaching the discussion session.

3) I think it will be appropriate to cite couple of key AFM-IR papers that are relevant for this study. For example, it will be worth mentioning that AFM-IR measurements on monolayers are not yet routine, and cite the very few works that have accomplished this task, for example:

- Chae et al. Nano Lett. 2017, 17, 9, 5587–5594

- Ruggeri et al. Nat. Comm. (2020) 11:2945

- ref #21 of the manuscript.

4) Critically, Fig 3 is inconsistent with the discussion and main conclusions of the manuscript and with the AFM-IR spectra.

- The on-figure labels are e) “1710 cm⁻¹”, f) “1730 cm⁻¹”, but the caption refers to the figure panels as e) as “1730 cm⁻¹” and f) as “1710 cm⁻¹” while the text (page 6): “The chemical mapping... fixed wavenumbers 1600 cm⁻¹ (Figure 3d), 1700 cm⁻¹ (Figure 3e), and 1730 cm⁻¹ (Figure 3f).

- The authors write (page 6):

“Specifically, within the chemical mapping at 1600 cm⁻¹ corresponding to the C=C bond stretching, the signal is virtually absent (in blue, Figure 3d) in the region attributed to the monolayer. Conversely, a strong signal is detected in the area corresponding to the multilayered network (illustrated in red, Figure 3d). For the band centered at 1730 cm⁻¹, the maximum signal intensity is observed in the monolayer zone, while for the 1700 cm⁻¹ band, the highest signal intensity is situated over the multilayer (as illustrated in Figures 3e and 3f).

— Critically, and contrary to the authors assertions, while a stronger intensity at 1730 cm⁻¹ should be expected for the monolayer based on the blue spectrum in fig. 3c, all the 3 AFM-IR absorption images (fig 3d, 3e, 3f) show stronger intensity in the multilayer regions than in the region attributed to the monolayer. Since panels d,e show zero intensity in the regions attributed to the monolayer, it is likely that the caption is incorrect and the on-figure label for panel-f and text correctly refer to this figure as the absorption map at 1730 cm⁻¹. However, even for this map the intensity on the multilayer island regions (orange, corresponding to 5-6 mV) shows stronger signal than for the monolayer regions (purple, about 3 mV). This is clearly inconsistent with the spectra in figure 3c.

— Since the C=O intensity and its peak-shift to 1740 cm⁻¹ are used in the paper to infer the absence of electronic conjugation with the phenyl ring in the monolayer regions and first layer of the multilayer regions this inconsistency undermines the authors conclusions. Please comment.

5) On Page 6 the authors write: “In our experimental setup, the incident light is polarized perpendicular to the HOPG surface.” I believe this may be incorrect as typically the IR laser light is incident at 20-30 degrees from the sample surface (i.e. has both vertical and horizontal polarization components). This may be different in the authors’ setup but was not specified.

6) The “5.6 nm” label in Fig 1c is hard to read. I would be better to change its color to black. The peak

labels in figure S3 are overlapped with the spectrum. Please rework the figure to avoid overlaps. (also, 2 decimal digits for the peak position labels are likely not necessary).

Besançon, October 3rd, 2023.

Dear reviewers,

we have carefully considered your different comments and suggestions. We have made significant modifications and improvements in the revised manuscript that address the different points raised by you. In the following, we are making point-by-point answers to your concerns. All significant changes in the manuscript have been highlighted in yellow in the revised version for your convenience.

Sincerely yours,

Dr. F. Chérioux

Reviewer #1 (Remarks to the Author):

Hamadeh et. al. reported the conformation assessment of molecular self-assemblies from 2D to 3D. The AFM work and data analyzing are complete, but the mechanism remains elusive. The combination of SPM and AFM-IR is unique, however the manuscript may need to re-organize in an easy-to-read fashion.

We thank the reviewer for her/his constructive comments.

1.If STM can evidence layered structures for this system?

No, it's nearly impossible to obtain STM images through layers that are several nanometers thick. That's why we studied multilayer systems using AFM. We used STM images to analyze the organization of monolayers in detail in order to propose the most comprehensive adsorption model possible.

2.If there is a strong computer simulation/calculation to support the mechanism?

No, we have not conducted numerical simulations of the infrared spectra. The comparison between the bulk spectrum and the multilayers is convincing. The shift in spectra with respect to conjugation is also a well-known phenomenon in the literature. We have not identified any barriers that could have been addressed through numerical simulations.

3.“This combined analytical approach can be universally applied to unravel the intricacies of on-surface synthesis processes”: I am confusing regarding this conclusion; I think there is no evidence to demonstrate this.

We agree with this comment, and the final sentence of the conclusion has been revised to remove this ambiguity.

Reviewer #2 (Remarks to the Author):

In “Toward Conformational Identification of Self-Assembled Molecules from 2D to 3D” Chérioux and colleagues, use atomic resolution STM and nanoscale resolution AFM-IR (an AFM-based analogue of IR spectroscopy) to study the intramolecular conformation of molecule-based supramolecular networks. The samples studied range from monolayers (characterized by a 2D arrangement/periodicity) and multilayers, which could in principle be considered 3D, as the authors did.

I like the manuscript approach which leverages the synergy (I believe for the first time) between atomic resolution STM and nanoscale resolution AFM-IR to evince the conformation of the biphenyl core and ester group in EsterC18 (octadecyl 4'-octadecyloxy-4-biphenylcarboxylate) on HOPG. The main claim of the paper is the qualitative determination of the conformation of the dihedral angle between the two aromatic groups in the biphenyl core and between one of the aromatic groups and the ester moiety.

Specifically, in the monolayer and in the first layer of a multilayer sample, the biphenyl group is mostly in plane while the ester group is rotated (weak electronic conjugation) while in the successive layers the biphenyl assumes a conformation more akin its bulk form (more rotated) and the dihedral angle with ester group is reduced to increase the electronic conjugation between the 2 chemical moieties. Overall, I find the manuscript novel, well written and of appropriate length. I also believe that it will appeal to a broad audience since measuring the conformation of molecules at the nanoscale is of interest in applications ranging from biology to molecular electronics. Technically, the manuscript is also relevant because although AFM-IR has recently increased in popularity, its application to very thin (i.e. monolayer samples) has been very limited due to the typical low signal intensity on such samples. While I'm generally enthusiastic about the manuscript, there are some inconsistencies between Figure 3, its caption, and the main text which contradict the main conclusions of the paper. (Perhaps the wrong figure 3 was used?) I would be happy to recommend the manuscript for publication if the authors can address this issue along with other minor suggestions detailed below.

We thank the reviewer for her/his constructive comments. In the following, we address the points raised by the reviewer.

1) I find the title and abstract a bit misleading as they seem to point to a sort of dynamic effect, while instead the "evolution of molecular conformation" refers (figuratively) to different samples, i.e. no evolution at all. Perhaps something like "assessing the conformation of molecules self-assembled in 2D and 3D" or something like that may be better.

We fully agree with this comment. There is no dynamic aspect in our study because we did not characterize the transformation from a 2D assembly to a 3D one, but rather two entirely distinct systems. As a result, we have modified the title and abstract to reflect this observation.

2) Since the molecules under study are characterized by many dihedral angles, i.e., conformations (both at the core and in the aliphatic chains) I think it would be useful to the reader to mention in the abstract what kind of molecules and which dihedral angles are analyzed. Otherwise, it is not clear what the main focus of the paper is before reaching the discussion session.

Thank you for your comment. We have added a sentence about dihedral angles to clarify the abstract.

3) I think it will be appropriate to cite couple of key AFM-IR papers that are relevant for this study. For example, it will be worth mentioning that AFM-IR measurements on monolayers are not yet routine, and cite the very few works that have accomplished this task, for example:

- Chae et al. *Nano Lett.* 2017, 17, 9, 5587–5594
- Ruggeri et al. *Nat. Comm.* (2020) 11:2945
- ref #21 of the manuscript.

These major references have been added in the main manuscript (new refs 22 and 23), and a sentence is added to comment the challenge of measuring this kind of sample with AFM-IR.

4) Critically, Fig 3 is inconsistent with the discussion and main conclusions of the manuscript and with the AFM-IR spectra.

- The on-figure labels are e) "1710 cm⁻¹", f) "1730 cm⁻¹", but the caption refers to the figure panels as e) as "1730 cm⁻¹" and f) as "1710 cm⁻¹" while the text (page 6): "The chemical mapping... fixed wavenumbers 1600 cm⁻¹ (Figure 3d), 1700 cm⁻¹ (Figure 3e), and 1730 cm⁻¹ (Figure 3f).

- The authors write (page 6):

“Specifically, within the chemical mapping at 1600 cm^{-1} corresponding to the C=C bond stretching, the signal is virtually absent (in blue, Figure 3d) in the region attributed to the monolayer. Conversely, a strong signal is detected in the area corresponding to the multilayered network (illustrated in red, Figure 3d). For the band centered at 1730 cm^{-1} , the maximum signal intensity is observed in the monolayer zone, while for the 1700 cm^{-1} band, the highest signal intensity is situated over the multilayer (as illustrated in Figures 3e and 3f).

□ Critically, and contrary to the authors assertions, while a stronger intensity at 1730 cm^{-1} should be expected for the monolayer based on the blue spectrum in fig. 3c, all the 3 AFM-IR absorption images (fig 3d, 3e, 3f) show stronger intensity in the multilayer regions than in the region attributed to the monolayer. Since panels d,e show zero intensity in the regions attributed to the monolayer, it is likely that the caption is incorrect and the on-figure label for panel-f and text correctly refer to this figure as the absorption map at 1730 cm^{-1} . However, even for this map the intensity on the multilayer island regions (orange, corresponding to 5-6 mV) shows stronger signal than for the monolayer regions (purple, about 3 mV). This is clearly inconsistent with the spectra in figure 3c.

□ Since the C=O intensity and its peak-shift to 1740 cm^{-1} are used in the paper to infer the absence of electronic conjugation with the phenyl ring in the monolayer regions and first layer of the multilayer regions this inconsistency undermines the authors conclusions. Please comment.

We would like to extend our apologies to the referee and express our gratitude for her/his feedback. Indeed, we acknowledge making significant errors in the creation of this figure. Specifically, in the submitted manuscript, the IR maps in Figures 3d and 3e were identical, and the Figure 3f was incorrect. Consequently, we have recreated the figure with the accurate maps.

However, the text was accurate. These glaring copy-paste errors account for the inconsistencies pointed out by the referee. We sincerely appreciate her/his exceptionally insightful analysis. Regarding the new maps in Figure 3e, the observed maximum for the 1710 cm⁻¹ band (red, around 3mV) indeed aligns with the multilayer region. In Figure 3f, the peak signal (red, 2 mV) is maximum over the monolayer. These mappings are consistent with the spectra presented in Figure 3b.

In this way, IR mappings and IR spectra are in perfect agreement. These experimental data serve to support our argument, which is that molecules adopt a different conformation when they are involved in a monolayer compared to a multilayer. Furthermore, the multilayer conformation is the same as that of the molecules in the bulk solid.

Once again, we apologize for these copy-paste errors that slipped through our multiple readings before the submission of this article.

5) On Page 6 the authors write: "In our experimental setup, the incident light is polarized perpendicular to the HOPG surface." I believe this may be incorrect as typically the IR laser light is incident at 20-30 degrees from the sample surface (i.e. has both vertical and horizontal polarization components). This may be different in the authors' setup but was not specified.

We agree with the reviewer that we made a short cut in our explanation about the polarization. We have replaced the sentence by a more complete explanation referring to our previous studies on polarization.

"In our setup the incident light has an angle about 30° from the surface and is polarized in the plane of incidence. Considering the gold coating of the tip and this configuration of illumination, we have already demonstrated that the electric field below the tip is dominated by its perpendicular component²⁶"

6) The "5.6 nm" label in Fig 1c is hard to read. I would be better to change its color to black. The peak labels in figure S3 are overlapped with the spectrum. Please rework the figure to avoid overlaps. (also, 2 decimal digits for the peak position labels are likely not necessary).

The two figures have been corrected.

REVIEWERS' COMMENTS:

Reviewer #2 (Remarks to the Author):

The authors have addressed all previous comment well and therefore I can recommend the manuscript for publication. Just a minor edit to correct typo on the new text (page 2) should be implemented as follows:

“However, it's worth noting that in the literature, **there are only reports of AFM-IR** studies conducted on few layered samples, specifically those with a thickness of a few nanometers²¹⁻²³.”

They probably meant “...there are only **a few** reports of AFM-IR studies...”